# Extensive Adaptive Variation in Gene Expression within and between Closely Related Horseshoe Bats (Chiroptera, *Rhinolophus*) Revealed by Three Organs

**DOI:** 10.3390/ani12233432

**Published:** 2022-12-06

**Authors:** Jun Li, Keping Sun, Wentao Dai, Haixia Leng, Aoqiang Li, Jiang Feng

**Affiliations:** 1Jilin Provincial Key Laboratory of Animal Resource Conservation and Utilization, Northeast Normal University, Changchun 130117, China; 2Key Laboratory of Vegetation Ecology, Ministry of Education, Changchun 130024, China; 3College of Life Science, Jilin Agricultural University, Changchun 130118, China

**Keywords:** adaptive expression variation, natural selection, closely related species, phenotype, bat

## Abstract

**Simple Summary:**

The role of natural selection on evolution at the gene expression level is still not well-understood. However, it is important for better understanding the molecular mechanisms of species differentiation. By identifying differentially expressed genes with variation forced by natural selection, we found that expression variation and variation related to phenotypic divergence were extensively adaptive between both species and subspecies for three closely related horseshoe bats. In addition, despite the immune-related adaptive expression variation found both within and between species, there were different functional patterns with respect to adaptive expression variation between two taxonomic scales as well. These results could be helpful for better comprehending the role of natural selection on evolution, and providing scientific basis for the protection of bat species.

**Abstract:**

In the process of species differentiation and adaption, the relative influence of natural selection on gene expression variation often remains unclear (especially its impact on phenotypic divergence). In this study, we used differentially expressed genes from brain, cochlea, and liver samples collected from two species of bats to determine the gene expression variation forced by natural selection when comparing at the interspecific (*Rhinolophus siamensis* and *R. episcopus episcopus*) and the intraspecific (*R. e. episcopus* and *R. episcopus* spp.) levels. In both cases, gene expression variation was extensively adaptive (>66.0%) and mainly governed by directional selection, followed by stabilizing selection, and finally balancing selection. The expression variation related to acoustic signals (resting frequency, RF) and body size (forearm length, FA) was also widely governed by natural selection (>69.1%). Different functional patterns of RF- or FA-related adaptive expression variation were found between the two comparisons, which manifested as abundant immune-related regulations between subspecies (indicating a relationship between immune response and phenotypic adaption). Our study verifies the extensive adaptive expression variation between both species and subspecies and provides insight into the effects of natural selection on species differentiation and adaptation as well as phenotypic divergence at the expression level.

## 1. Introduction

The importance of variation in gene expression for speciation and differentiation has been confirmed in many studies [1,2,3,4]. The expression variation forced by neutral drift or natural selection, which can be considered neutral or adaptive variation, has attracted attention in recent years [5,6,7]. Identifying the neutral or adaptive variation in gene expression between taxa and uncovering the relative influences of natural selection versus neutral drift can help clarify the characteristics of evolution during species differentiation and adaptation at the gene expression level. Several studies have focused on gene expression evolved under positive selection or neutral drift [8,9]. For example, early research on yeast and primates suggested that much of the extensive variation in interspecific gene expression was neutral [10,11]. Extensive hallmarks of positive selection were subsequently identified among population for certain non-model species, such as *Fundulus heteroclitus* [12].

By observing the variation in gene expression among individuals within a taxon and that between taxa, genes can show different patterns when their variation is governed by directional selection, stabilizing selection, or balancing selection. Using this method, several studies have determined the fraction of genes with the expression variation forced by natural selection. For instance, the expression variation of differentially expressed genes (DEGs) likely governed by directional selection and stabilizing selection were identified for closely related species within *Oryza*, and they found that the degrees of expression variation forced by positive selection were up to 60% and similar among different tissues [5]. However, the relative contributions of directional selection, stabilizing selection, or balancing selection to expression divergence remain unclear, especially in different organs of mammals.

Furthermore, neutral drift and natural selection can govern the variation of phenotypic characteristics. Many studies have suggested that variation in gene expression is the main cause of the phenotypic differentiation of closely related species [13,14]. For example, a series of studies on European black crows, gray crows, and hybrids have indicated that the rapid differentiation of phenotype is associated with extensive variation in gene expression [4,15,16]. Most studies have been limited to explore the expression variation related to the divergence of characteristic phenotypes by screening the trait-related differentially expressed genes and enrichment analysis [17,18]. Relatively little is known about the evolutionary process of the expression variation related to the differentiation and adaptation of phenotypes.

*Rhinolophus siamensis* and *R. episcopus* provide an opportunity to study the evolutionary forces of variation in gene expression. These closely related bat species diverged ca. 1.51 Ma [19]. A previous study detected the gene expression diversity of three organs (brain, cochlea, and liver) in interspecific (*R*. *siamensis* and *R. e. episcopus*) and intraspecific (*R. e. episcopus* and *R. episcopus* spp.) comparisons [20]. A degree of phenotypic divergence, including the dominant frequency of echolocation calls and body size, was found in both interspecific and intraspecific comparisons among these taxa [21]. For bats, echolocation calls are closely associated with many life activities, such as communication, courtship, foraging, navigation, individual recognition, and territory defense. Echolocation calls are therefore considered to be one of the most important sensory characteristics of bats and have great significance in the differentiation and adaptation of bat species [22,23,24]. Furthermore, forearm length (FA), as a feature that influences overall wing and body size, is an important standard morphological index for bats [25]. A previous study showed that lots of DEGs related to FA were detected among different *R. ferrumequinum* populations [17], suggesting the effect of body size on gene expression differences.

In this study, to clarify the evolutionary processes of gene expression variation within and between species, we used the DEGs obtained from brain (vital organ responsible for organism-level regulation), cochlea (receiving organ of echolocation), and liver (primary metabolic organ) samples to conduct inter-specific and inter-subspecific comparisons. We propose that the expression variation, especially related to phenotypic divergence, between taxa is adaptive to some extent. Therefore, we identified the DEGs and phenotype-related DEGs with expression variation forced by directional selection, stabilizing selection, and balancing selection, and assessed the extent of natural selection acting on the expression variation. We are also interested in asking whether the role of natural selection for expression variation were different (i) among three tested organs, and (ii) between inter-specific and inter-subspecific comparisons. Furthermore, we investigated the function patterns of adaptive expression variation and those related to phenotypes.

## 2. Materials and Methods

### 2.1. Sample Collection and Data Acquisition

Brain, cochlea, and liver samples of *R*. *siamensis* (*n =* 4) and *R. e. episcopus* (*n =* 5) were collected from Hunan Province of China, and *R. episcopus* spp. (*n =* 5) collected from Yunan Province of China. We identified these taxa based on a previous study by Liu et al. [19], and the vouchers were deposited in Northeast Normal University, China. Considering that resting frequencies (RFs) of horseshoe bats are widely used to characterize echolocation vocalizations in previous studies [26,27], RFs of the dominant second harmonic of the constant frequency component were measured from fourteen individuals of three taxa. RNA sequencing data were generated from 42 samples, and the raw reads of all samples in this study were submitted to NCBI (Submission ID: SUB9705561; BioProject ID: PRJNA781455). DEGs (Appendix A) and RF-related DEGs (Appendix A) obtained in inter-specific and inter-subspecific comparisons for brain, cochlea, and liver samples in a previous study by this research group [20] were used in this study. In addition, FA, as the dominant external morphological characteristic of bats, was measured using digital calipers (0.01 mm). Based on a series of correlation analyses (Appendix A), FA-related DEGs were distinguished using the same methods used for the identification of RF-related DEGs. All sampling procedures were approved by the National Animal Research Authority in Northeast Normal University, China (approval number: NENU–20080416).

### 2.2. Expression Variation between Taxa and among Individuals within Taxon

Although DEGs could be directly screened as genes with high variation between taxa, further screening was still needed to reduce false positives. The number of fragments per kilobase of transcript sequence per millions of base pairs sequenced (FPKM) was used to quantify gene expression levels [28]. We performed zFPKM to normalize FPKM values [29] using R (version 3.6.0), and the code was presented in Appendix A. Based on the zFPKM of each DEGs, the analysis of variance (ANOVA) was used to determine the variance between taxa. DEGs with *p* values < 0.05 were considered genes with high expression variation between taxa.

In addition, we evaluated the expression variation among individuals within each taxon. The difference values of zFPKM were used to represent the expression divergence between individuals. The degree of the dispersion of expression divergence between individuals (standard deviation; SD values) was computed to evaluate the expression variation between individuals within each taxon for each gene. Furthermore, a ranking-based method was used to determine the threshold for genes with high variation among individuals within each taxon [6,30]. For each taxon, the distribution of ranked expression variation between individuals was conducted using all expressed genes. Two straight lines were further fitted to two linear parts of the distribution using the top 1% genes and bottom 80% genes [5]. The intersection point of these two straight lines was set as the critical point of high variation between individuals within taxon. The schematic diagram defining genes with small or large expression variation among individuals within taxon using rank-based method was shown in Appendix A.

### 2.3. Identifying Expression Variation Forced by Natural Selection, Especially Related to Phenotypic Divergence

Based on the definitions of expression variation between taxa and among individuals within each taxon, DEGs with low expression variation within either taxon but high variation between taxa were identified that the expression likely evolved under directional selection, and DEGs with low variation within either taxon and between taxa were identified that the expression likely evolved under stabilizing selection. An ANOSIM was conducted to evaluate the magnitude of the relationship between the expression variation within and between taxa using vegan package in R [31]. The DEGs with high variation within both taxa, low variation between taxa, and higher variation within than between taxa were identified that the expression likely evolved under balancing selection.

Based on identified RF- and FA-related DEGs (Appendix A), intersection analysis was performed to screen RF- or FA-related DEGs whose expression likely evolved under directional selection, stabilizing selection, or balancing selection.

### 2.4. Functional Enrichment Analysis

For DEGs whose expression likely evolved under natural selection, including directional selection, stabilizing selection, and balancing selection, GO and KEGG enrichment analyses were performed using *Pteropus alecto* (the most closely related species to *Rhinolophus* in the enrichment database) as the reference organism in each comparison for each organ. The enrichment analysis was also executed for RF- and FA-related DEGs whose expression was deemed to have evolved under natural selection. All enrichment analyses were performed in the STRING program (11.0) (https://string-db.org/, accessed on 7 March 2022) using ‘Multiple Proteins by Names/Identifiers’ platform, and FDR (corrected *p* value) < 0.05 was set as the enrichment significance threshold [32].

## 3. Results

### 3.1. Differences in Gene Expression within and between Taxa

In this study, 70.0% (453/647), 66.9% (715/1069), and 69.7% (482/692) of DEGs showed high variance between species for brain, cochlea, and liver, respectively. However, extensive high variance (>60% of DEGs) was only found for liver samples between subspecies (Appendix A).

According to the ranking-based distribution, gene expression varied extensively among individuals within each taxon (Appendix A). Most of the DEGs had little variable expression among individuals within each taxon for brain (inter-specific comparison: 79.6%, 515/647; inter-subspecific comparison: 69.9%, 369/528), cochlea (66.7%, 713/1069; 64.8%, 394/608), and liver (91.5%, 633/692; 97.1%, 1328/1368) samples. Only a small number of DEGs showed high variation among individuals within both taxa in each comparison (Appendix A).

As shown in the results of the analysis of similarities (ANOSIM) (Appendix A), the vast majority of DEGs showed higher variation between taxa than within taxon for three organs in both inter-specific (brain: 85.2%, 551/647; cochlea: 85.5%, 914/1069; and liver: 84.4%, 584/692) and inter-subspecific (brain: 85.6%, 452/528; cochlea: 90.6%, 551/608; and liver: 78.1%, 1068/1368) comparisons.

### 3.2. Expression Variation Extensively Governed by Natural Selection

The expression variation of a large number of DEGs were likely forced by directional selection, followed by stabilizing selection, while balancing selection was the least common for three organs in both inter-specific and inter-subspecific comparisons (Table 1). The expression variation of 39 and 52 DEGs were likely under natural selection for all three organs in the inter-specific and inter-subspecific comparisons, respectively (Figure 1a,b).

For brain, cochlea, and liver samples, the expression variation of most DEGs was likely forced by natural selection in both inter-specific (79.8%, 67.4%, and 91.6%, respectively) and inter-subspecific (70.3%, 66.0%, and 97.1%, respectively) comparisons. As shown in Figure 2, the proportion of DEGs whose variation were likely forced by directional selection was the largest (inter-specific comparison: 50.7–68.4%; inter-subspecific comparison: 47.2–70.3%), followed by stabilizing selection (inter-specific comparison: 16.0–23.1%; inter-subspecific comparison: 11.2–26.8%), and only a small proportion of that was likely subjected to balancing selection (inter-specific comparison: 0.2–0.7%; inter-subspecific comparison: 0.0–1.2%).

### 3.3. Functional Patterns of Adaptive Expression Variation

The enrichment results using DEGs whose variation were governed by natural selection were different between two comparisons. In the inter-specific comparison, many enriched gene ontology (GO) terms were related to cellular components and biological processes, including ‘organelle’, ‘cell cycle’, ‘chromosome’, ‘immune system’, and ‘stress and defense response’ for cochlea samples. It was significantly enriched in GO categories related to ‘extracellular region’ and ‘anchored component of membrane’ for brain samples, and ‘extracellular region and space’ for liver samples (Figure 3). In the inter-subspecific comparison, enriched GO terms were related to biological processes, including ‘cellular component movement and motility’, ‘cell migration’, and ‘the positive or negative regulation of immune process’ and ‘adaptive stimulus response’ for all three organs studied. The enriched functions also included’ cell communication’, ‘locomotion’, ‘signaling receptor’, and ‘signal transduction’ for cochlea and liver samples. In addition, several terms related to molecular functions varied among three organs, such as ‘calcium ion’, ‘glycosaminoglycan’, and ‘cell adhesion molecule binding’ for brain samples, ‘signaling and immune receptor activities’ for cochlea samples, and ‘immune receptor activity’ and ‘actin and protein binding’ for liver samples. Several terms related to cellular components were also enriched, including ‘extracellular region and space’ for all three organs, and ‘membrane and cell surface’ for cochlea and liver samples (Appendix A).

In the inter-specific comparison, only one enriched Kyoto Encyclopedia of Genes and Genomes (KEGG) pathway, ‘complement and coagulation cascades’, was found for brain samples. In the inter-subspecific comparison, many pathways related to ‘disease and immunoregulation’ were enriched for all three organs. Furthermore, pathways related to ‘regulation of actin cytoskeleton’ and ‘neuroactive ligand-receptor interaction’ were also enriched for brain and cochlea samples, respectively, while pathways related to ‘osteoclast differentiation and metabolism’ were enriched for liver samples (Appendix A).

### 3.4. Adaptive Expression Variation Related to Phenotypes

A number of DEGs with adaptive expression variation were related to resting frequency (RF) or forearm length (FA) in two comparisons for three organs (Table 2). For each organ in each comparison, the number of trait-related DEGs whose expression variation were governed by directional selection was the largest, followed by stabilizing selection. None of RF- or FA-related DEG with variation forced by balancing selection was found. Among 39 DEGs with adaptive expression variation for all three organs in the inter-specific comparison, 12 of them (*ZNF132*, *TMA7*, *MRNIP*, *PLIN1*, *DUSP22*, *MCMDC2*, *RBP2*, *FOLR2*, *RDH5*, *NRN1L*, *RAD51C*, and *SPATC1*) were related to both RF and FA (Appendix A). In the inter-subspecific comparison, the expression variation of 22 DEGs (*SIT1*, *RPL38*, *SASH3*, *CYTH4*, *DOCK2*, *INPP5D*, *CD300A*, *WAS*, *PTPN7*, *PTPN6*, *NCKAP1L*, *LYZ*, *ITGB2*, *HCLS1*, *LAIR1*, *TIMP1*, *FGD3*, *SLC15A3*, *CSF2RA*, *ARHGAP4*, *C1QA*, and *PTER*) were related to RF or FA was forced by directional selection (Appendix A).

For brain, cochlea, and liver samples, 85.2% (104/122), 78.7% (285/362), and 93.9% (139/148) of RF-related DEGs, respectively, and 81.0% (166/205), 78.1% (278/356), and 93.3% (126/135) of FA-related DEGs, respectively, exhibited the expression variation likely forced by natural selection in the inter-specific comparison. For brain, cochlea, and liver samples, higher proportions of RF-related (90.5%, 57/63; 78.9%, 97/123; 97.2%, 138/142) and FA-related (86.4%, 57/66; 79.5%, 97/122; 98.4%, 187/190) DEGs with adaptive variation were screened in the inter-subspecific comparison. The expression variation of RF- and FA-related DEGs governed by directional selection accounted for the highest proportion in each comparison (Figure 4).

For RF- and FA-related DEGs with adaptive variation, enriched GO terms and KEGG pathways were only obtained for cochlea samples and differed between two comparisons. For both RF- and FA-related DEGs with adaptive variation, enriched GO terms were widely related to ‘cell cycle process’, ‘chromosome’, and ‘DNA’ in the inter-specific comparison, while ‘immune process and regulation’ dominated in the inter-subspecific comparison (Appendix A). The KEGG pathways related to ‘cell cycle’ were enriched for both RF- and FA-related DEGs with adaptive variation in the inter-specific comparison. In the inter-subspecific comparison, significantly enriched pathways were related to ‘B cell receptor signaling pathway’ and ‘Leishmaniasis’ for RF-related DEGs with adaptive variation, and ‘B and T cell receptor signaling pathway’ and ‘primary immunodeficiency’ for FA-related DEGs whose variation were forced by natural selection.

## 4. Discussion

### 4.1. Extensive Adaptive Variation in Inter-Specific and Inter-Subspecific Gene Expression

This study investigated closely related horseshoe bats and obtained evidence that expression variation was governed by natural selection. Between both species and subspecies, expression variation was mainly governed by directional selection (47.2–70.3%), followed by stabilizing selection (11.2–26.8%), and finally balancing selection (<1.2%) (Figure 2).

Different degrees of adaptive expression variation were found for different taxa and even among different tissues in some previous studies. For instance, 10–17%, 10–30%, and more than 60% of expression variation were likely forced by natural selection for yeast, primates, and rice, respectively [5,6,33,34]. In this study, between both species and subspecies, the lowest expression variation governed by natural selection was more than 60%, which was really higher than other model systems [9,11,35], indicating extensive adaptive variation in gene expression during the process of differentiation and adaptation for these three closely related horseshoe bats.

### 4.2. Organ-Specificity of Expression Variation Governed by Natural Selection

Between both species and subspecies, the proportions of expression variation forced by natural selection were different among three organs. This suggested the organ-specific adaptive variation in gene expression. Furthermore, compared to brain and cochlea tissues, the proportion of adaptive expression variation was the highest for liver samples (>91.618%). This may be due to the fact that the liver, which is the key metabolic organ, has more interaction with the environment [36]. As a contrast, a large number of DEGs with adaptive expression variation were found for both brain and cochlea samples, and the proportion of adaptive expression variation was similar between these two organs. This finding further suggests that there may be a correlation in the regulation in brain and cochlea tissues at the expression level for bat species.

In both inter-specific and inter-subspecific comparisons, the enrichment functions varied among three organs. First, the organ-specific GO terms using DEGs with adaptive expression variation between species were manifested in cochlea tissues (Figure 3). Second, the adaptive expression variation related to molecular functions between subspecies was also specific for cochlea (Appendix A). These findings indicate that cochlea tissues show specific functional patterns of adaptive expression variation compared to the other two organs, which may be due to the fact that the cochlea, as the organ receiving echolocation calls, is unique in function for bats [37].

### 4.3. Functional Patterns of Adaptive Expression Variation at Two Different Classification Categories

The adaptive expression variation associated with disease and immune processes was identified in both inter-specific and inter-subspecific comparisons. The enriched terms or pathways related to the immune system, and complement and coagulation cascades, which are associated with many immune deficiency diseases [38], in the inter-specific comparison, combined with those related to immune receptor activities and immunoregulation in the inter-subspecific comparison. This means that there are extensive adaptive expression variation related to immune processes during the differentiation and adaptation of these three closely related horseshoe bats.

Compared to the inter-specific comparison, the enrichment results in the inter-subspecific comparison showed unique functional patterns for adaptive expression variation. These terms and pathways indicated the adaptive response associated with the movement and motility of the cellular component at the expression level. Adaptive expression variation related to calcium ion, membrane and cell surface, glycosaminoglycan, signaling receptor and signal transduction, cell adhesion molecule binding, and neuroactive ligand-receptor interaction was also found between subspecies. Calcium plays an important role in neurodegeneration, memory formation, and the transformation of various information [39,40,41]. Several studies have demonstrated that calcium is also a key signaling ion involved in many different intracellular and extracellular processes, including the functions of the nervous system (ranging from neurotransmitter release to synaptic activities), cell-cell communication, and adhesion [42]. Cell adhesion molecules are positioned to play pivotal roles in the migration, target penetration, and synapse formation of sensory neurons [43]. As to glycosaminoglycans, they can organize the extracellular matrix, contribute to cell-matrix interactions, and regulate cell signaling [44]. The adaptive expression variation related to growth and development, including the regulations of actin cytoskeleton, actin and protein binding, and osteoclast differentiation and metabolism, may be due to the divergence of body size between two subspecies.

### 4.4. Trait-Related Expression Variation Forced by Natural Selection

As the phenotypic parameters related to numerous life activities of bats, RF and FA were selected to explore and elucidate the role of adaptive expression variation associated with phenotypic differentiation and adaptation. Between species or subspecies, the expression variation related to acoustic signal and body size was widely governed by natural selection (>69.1%), and the degrees of adaptive expression variation were highest for the liver, followed by the brain, and lowest in the cochlea (Figure 4). This organ-specificity was consistent with the results of the adaptive variation in gene expression. In addition, the adaptive expression variation related to acoustic signal and body size was primarily forced by directional selection, followed by stabilizing selection, with no evidence of balancing selection.

Among DEGs whose variation was forced by directional selection and related to the divergence of RF between species (Appendix A), *RBP2* and *RDH5* were associated with visual sense [45,46]. Although previous studies have suggested that the vision of echolocation bats inhabiting in caves has degenerated, several studies have found evidence that visual sense still plays an important role in the living activities of these bats [47,48,49]. For example, a study on *Rousettus aegyptiacus* found that there was a switch between vision and echolocation, such as using visual sense to determine the flight direction, and using echolocation calls to identify obstacles [50]. The results of the present study may indicate an adaptive combination of visual and auditory sensory regulation at the expression level during the species differentiation of these three *Rhinolophine*. However, *CYTH4* and *INPP5D*, two RF-related DEGs with expression variation forced by directional selection in the inter-subspecific comparison, were associated with the nervous system [51,52], indicating a potential association between the regulation of neural activity and the adaption of acoustic signals in bats (Appendix A). This association has been proposed in many previous studies on the differentiation of acoustic signals for bats [17,53].

The enrichment results revealed some different functional patterns of RF- or FA-related expression variation governed by natural selection between the two comparisons. It was mainly manifested as the abundant adaptive trait-related expression variation associated with immunologic process, which was just found between subspecies (Appendix A). Certain immune-related DEGs, such as *SASH3* [54], *CD300A* [55], *WAS* [56], *NCKAP1L* [57], *HCLS1* [58], *LAIR1* [59], *SLC15A3* [60], and *C1QA* [61], exhibited adaptive expression variation between subspecies and were associated with RF or FA (Appendix A). These results indicated an attractive relationship between immune response and the adaption of acoustic signals and body size in bats. Many studies have suggested that resources invested in reproduction often come at the expense of the ability to mount an immune response [62]. The importance of acoustic signals for reproduction has been demonstrated in many species [63,64,65]. Research on sagebrush crickets has shown that males are unable to sustain the costly acoustic signal needed to attract additional females due to triggering an energetically costly immune response [62]. Furthermore, studies on bats have demonstrated that the frequency of echolocation calls can be affected by FA [66], which explains the consistency of functional patterns of expression variation related to RF and FA.

## 5. Conclusions

In this study, the relative influences of natural selection on expression variation were evaluated.at both inter-specific and inter-subspecific levels for three organs (cochlea, brain and liver) from two closely related *Rhinolophus*. We found the expression variation and those related to phenotypic divergence were extensive adaptive, and the proportion of expression variation governed by directional selection was the highest. The relative influences of natural selection on expression variation was organ-specific, but similar between two taxonomic levels. The functional patterns of adaptive expression variation included immune regulation at both two taxonomic levels, and it also included ion activity and signal transduction, nervous system, and growth and development at subspecific level. This implied that there were differentiation and adaptation related to these functions mentioned above at the expression level in the process of evolution for three closely related horseshoe bats. Furthermore, the functional patterns of adaptive expression variation associated with phenotypes were different between two taxonomic levels, manifested as cellular processes at species level and immunoregulation at subspecies level. These results could be helpful to better understand the role of natural selection on the differentiation and adaptation of species at the gene expression level and provide scientific basis for the protection of bat species.

## Figures and Tables

**Figure 1 animals-12-03432-f001:**
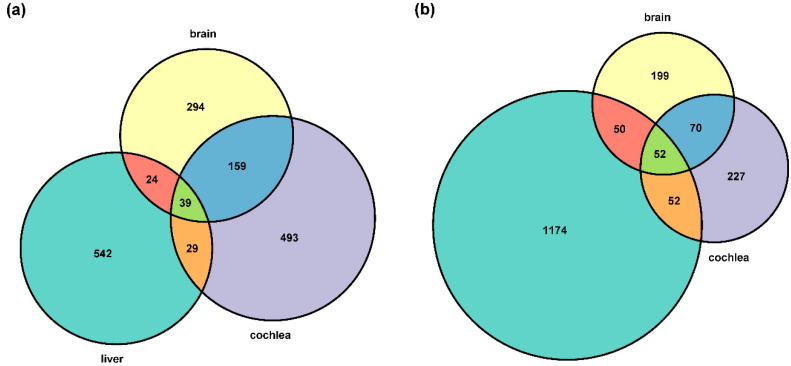
Venn diagram showing the numbers of intersecting differentially expressed genes (DEGs) under natural selection among the three organs in (**a**) the inter-specific and (**b**) inter-subspecific comparison. The labels ‘cochlea’, ‘brain’, and ‘liver’ indicate the cochlea, brain, and liver samples in each comparison, respectively. The size of diagram is proportional to quantity. Colors have no meaning.

**Figure 2 animals-12-03432-f002:**
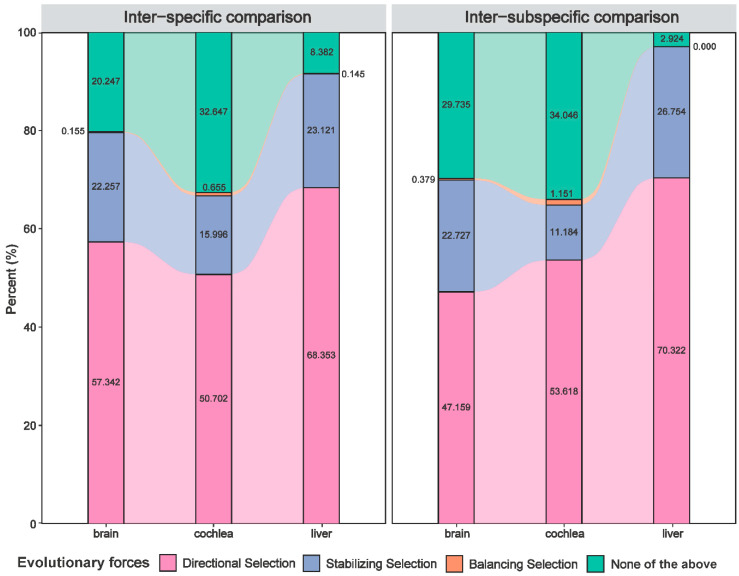
The ratio of differentially expressed genes (DEGs) whose expression likely evolved under natural selection, including directional selection, stabilizing selection, and balancing selection, in the inter-specific and inter-subspecific comparisons for three organs.

**Figure 3 animals-12-03432-f003:**
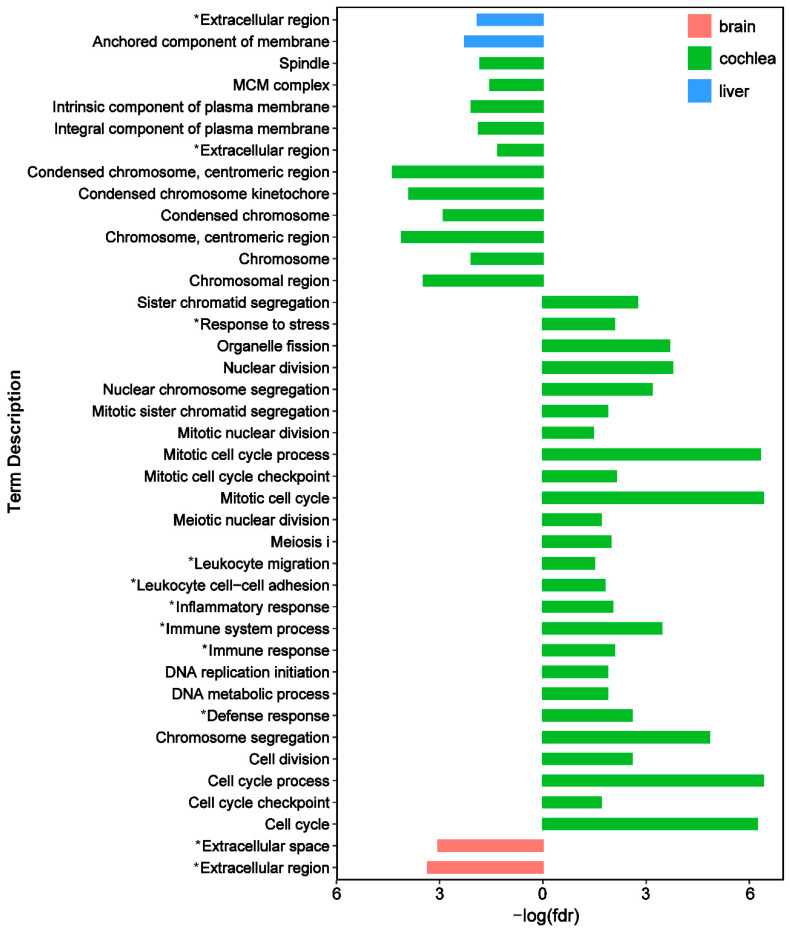
Significantly enriched gene ontology (GO) terms obtained using differentially expressed genes (DEGs) whose expression likely evolved under natural selection in the inter-specific comparison for three organs. Left and right indicate terms related to cellular components and biological processes, respectively. The bar colors represent different organs. ‘*’ indicates the terms also enriched in the inter-subspecific comparison.

**Figure 4 animals-12-03432-f004:**
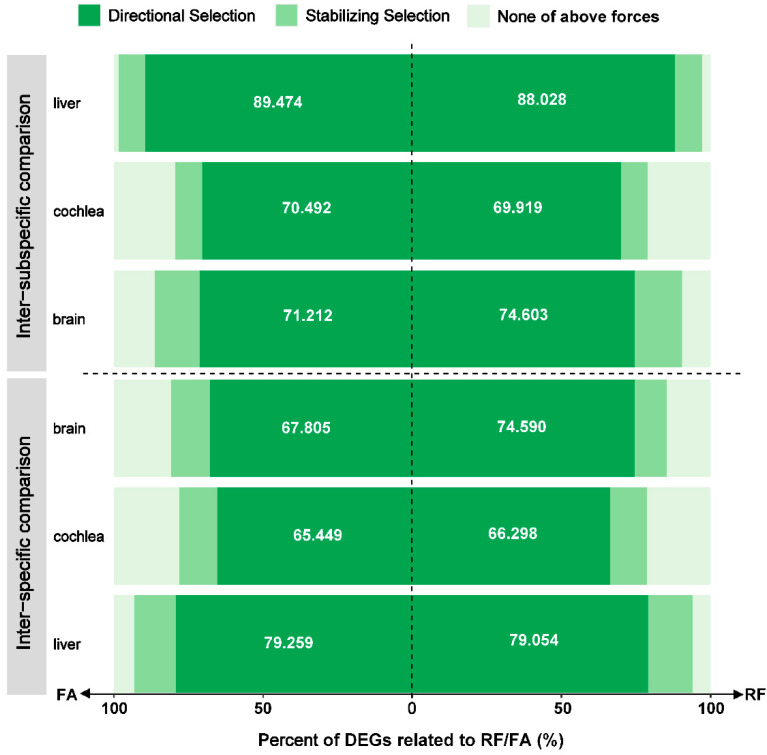
The ratio of trait-related differentially expressed genes (DEGs) whose expression likely evolved under natural selection, including directional selection and stabilizing selection, in the inter-specific and inter-subspecific comparisons for three organs.

**Table 1 animals-12-03432-t001:** Numbers of differentially expressed genes (DEGs) with expression variation forced by natural selection in the inter-specific and inter-subspecific comparisons for three organs.

Comparison	Organ	Number of DEGs			
		Directional Selection	Stabilizing Selection	Balancing Selection	In Total
Inter-specific comparison	brain	371	144	1	516
	cochlea	542	171	7	720
	liver	473	160	1	634
Inter-subspecific comparison	brain	249	120	2	371
	cochlea	326	68	7	401
	liver	962	366	0	1328

**Table 2 animals-12-03432-t002:** Numbers of differentially expressed genes (DEGs) with expression variation related to traits and forced by natural selection in the inter-specific and inter-subspecific comparisons for three organs. RF and FA represent the resting frequency of echolocation calls and forearm length, respectively.

Comparison	Organ	Number of DEGs Related to RF/FA			
		Directional Selection	Stabilizing Selection	Balancing Selection	In Total
Inter-specific comparison	brain	91/139	13/27	0/0	104/166
	cochlea	240/233	45/45	0/0	285/278
	liver	117/107	22/19	0/0	139/126
Inter-subspecific comparison	brain	47/47	10/10	0/0	57/57
	cochlea	86/86	11/11	0/0	97/97
	liver	125/170	13/17	0/0	138/187

## Data Availability

The raw data of 42 samples used in transcriptome sequencing in this article are available in GenBank Nucleotide Database at https://www.ncbi.nlm.nih.gov/bioproject/PRJNA781455, accessed on 18 November 2021, and can be accessed with the Submission ID SUB9705561 and BioProject ID PRJNA781455.

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
