# Peer review of "Extensive Adaptive Variation in Gene Expression within and between Closely Related Horseshoe Bats (Chiroptera, *Rhinolophus*) Revealed by Three Organs"

_animals, 2022, doi:10.3390/ani12233432_

Round 1

Reviewer 1 Report

Interesting manuscript....  I have a few comments:

1) The authors try and use body size and other morphometric differences between species and subspecies as a way of explaining differences in expression levels. If this approach is to be taken, the authors should lay out a reason for selecting these taxa as part of their experimental design.  

2) How do we know the taxa are identified correctly?  Where are vouchers deposited?  Who performed the identifications?

3) What was the variation between individuals within a taxon?  How did those data compare to between taxon comparisons?

4) Explain the what 66% 69% means relative to overall variation from other studies.  It is about average?  Really high, really low?  We need some benchmarks for context....

5) The vision arguments really a strawman... Most Pteropodid bats have good vision, in fact many are diurnal.  Only Rousettes, which live in caves, use echolocation and it is different than in other bat group.... So this argument needs to be better framed.

6) Need better hypotheses....

Author Response

Response to Reviewer 1 Comments

Interesting manuscript.... I have a few comments:

Point 1: The authors try and use body size and other morphometric differences between species and subspecies as a way of explaining differences in expression levels. If this approach is to be taken, the authors should lay out a reason for selecting these taxa as part of their experimental design.

Response 1: Thaks for your suggestion. In order to indicate why we used body size, such as forearm length, as a way of explaining differences in expression levels, we added one sentence at the end of this paragragh. The added content is “Previous study showed that lots of DEGs related to FA were detected among different R. ferrumequinum populations (Zhao et al., 2019), suggesting the effect of body size on gene expression differences.” in lines 89-91.

Point 2: How do we know the taxa are identified correctly? Where are vouchers deposited? Who performed the identifications?

Response 2: We identified the taxa based on a previous study (Liu et al., 2019), which clarified the classification of these taxa based on morphological, acoustic and molecular data. The vouchers were deposited in Northeast Normal University, China. We added these information to Section 2.1 in lines 115-116.

Reference:

Liu, T.; Sun, K.P.; Csorba, G.; et al. Species delimitation and evolutionary reconstruction within an integrative taxonomic framework: A case study on Rhinolophus macrotis complex (Chiroptera: Rhinolophidae). Mol. Phylogen. Evol. 2019, 139, 106544.

Point 3: What was the variation between individuals within a taxon? How did those data compare to between taxon comparisons?

Response 3: We evaluated the expression variation among individuals within each taxon, which was different from the expression variation between taxa. To clarify the difference, we added “In addition, we evaluated the expression variation among individuals within each taxon.” before “The difference values of zFPKM were used to represent the expression divergence between individuals.” in lines 140-141. And we used a schematic diagram to better explain the expression variation among individuals within each taxon, as shown in Figure_S1, which defined the genes with small or large expression variation among individuals. We also revised the legend of Figure_S1 for clarifing its meaning clearly (lines 427-431).

Figure_S1: Schematic diagram defining genes with small or large expression variation among individuals within taxon using rank-based method. Gray lines are two fitted straight lines using first 1% and bottom 80% genes, respectively, and the intersection point is the cut-off point between small and large expression variation among individuals with taxon (The rank-based method is cited from Gilad et al. 2006, Blekhman et al. 2008, Guo et al. 2016).

Point 4: Explain the what 66% 69% means relative to overall variation from other studies. It is about average? Really high, really low? We need some benchmarks for context....

Response 4: Thanks for your suggestion. In Section 4.1, we revised the sentence as “In this study, between both species and subspecies, the lowest expression variation governed by natural selection was more than 60%, which was really higher than other model systems, indicating extensive adaptive variation in gene expression during the process of differentiation and adaptation for these three closely related horseshoe bats.” in lines 311-316.

Point 5: The vision arguments really a strawman... Most Pteropodid bats have good vision, in fact many are diurnal. Only Rousettes, which live in caves, use echolocation and it is different than in other bat group.... So this argument needs to be better framed.

Response 5: Thanks for your comment. We stated the vision arguments for echolocation bats inhabiting in caves. The revised sentence is “Although previous studies have suggested that the vision of echolocation bats inhabiting in caves has degenerated, several studies have found the evidence that visual sense still plays an important role in the living activities of these bats.” in lines 378-380.

Point 6: Need better hypotheses....

Response 6: Thanks for your comment. We revised the hypotheses as “We propose that the expression variation, especially related to phenotypic divergence, between taxa is adaptive to some extent. Therefore, we identified the DEGs and phenotype-related DEGs with expression variation forced by directional selection, stabilizing selection, and balancing selection, and assessed the extent of natural selection acting on the expression variation. We are also interested in asking whether the role of natural selection for expression variation were different (i) among three tested organs, and (ii) between inter-specific and inter-subspecific comparisons. Furthermore, we investigated the function patterns of adaptive expression variation and those related to phenotype.” (lines 96-110).

Reviewer 2 Report

Except for the title, which should include the second objective that they propose. I find a very original, different and very well written work.

Author Response

Response to Reviewer 2 Comments

Point 1: Except for the title, which should include the second objective that they propose. I find a very original, different and very well written work.

Response 1: Thanks very much for your affirmation. To contain all the objectives, we revised the title to “Extensive Adaptive Variation in Gene Expression within and between Closely Related Horseshoe Bats (Chiroptera, Rhinolophus) Revealed by Three Organs”.

Reviewer 3 Report

This is a very interesting and well composed study about the role of natural selection on gene expression in horseshoe bats. The samples are from wild caught bats and the authors look at three types of selection based on various phenotypic traits. It reads excellently and I wonder if I am reading a second or third version of the manuscript. It is a pleasure to read.

The abstracts both adequately describe the study, but some small changes

Line 21 - delete the before evolution

Line 27 - I assume you mean INTRA specific here - I suppose you can say intrasubspecific but you are comparing two species and two subspecies...so intraspecific seems a better option

The key words can be more diverse as they repeat words in the title. What about phylogeny? Phenotype?

Introduction - excellent first three paragraphs putting the work in a broader context and making it suitable for the journal and for others to cite not working on bats

Line 73 - can make the second Rhinolophus R.

Line 75 - delete in before circa

Line 77 - I think you mean intra again - same with line 79

It is interesting to consider echolocation itself the phenotype - wouldn't it be better the suite of characters contributing to echolocation?  The cochlea does make sense here as one representative

Line 108 - intra again

Section 2.i - need to mention ethics on the bat collection and their conservation and protection status

Line 162 - a very high level of precision - would suggest to round to one decimal point or to justify this precision - I can see that the differences are not always great but they still do not overlap even at one decimal

Figure 1 - does the overlap have a meaning? Is it showing it at a scale?

I am really impressed with the discussion. It is well ordered, puts the work well in the context of the literature, and I find the discussion on the costly immune response of different types of calling in different sexes fascinating and would apply this to my own work. A huge array of references are used putting this work in a broad context.

Author Response

Response to Reviewer 3 Comments

Point 1: This is a very interesting and well composed study about the role of natural selection on gene expression in horseshoe bats. The samples are from wild caught bats and the authors look at three types of selection based on various phenotypic traits. It reads excellently and I wonder if I am reading a second or third version of the manuscript. It is a pleasure to read.

Response 1: We are appreciated you for all of the constructive suggestions. According to your comments, we have modified our manuscript as below.

Point 2: The abstracts both adequately describe the study, but some small changes

Line 21 - delete the before evolution

Line 27 - I assume you mean INTRA specific here - I suppose you can say intrasubspecific but you are comparing two species and two subspecies...so intraspecific seems a better option

Response 2: Based on your suggestion, we deleted “the” before “evolution” in line 21, and revised “inter-subspecific” to “intraspecific” in line 27.

Point 3: The key words can be more diverse as they repeat words in the title. What about phylogeny? Phenotype?

Response 3: Thanks for your suggestion. We revised the key words as “adaptive expression variation; natural selection; closely related species; phenotype; bat” (line 38).

Point 4: Introduction - excellent first three paragraphs putting the work in a broader context and making it suitable for the journal and for others to cite not working on bats

Line 73 - can make the second Rhinolophus R.

Line 75 - delete in before circa

Line 77 - I think you mean intra again - same with line 79

Response 4: Based on your suggestion, we revised “Rhinolophus episcopus” to “R. episcopus” in line 75, deleted “in” in line 77, and changed “inter-subspecific” to “intraspecific” in line 79 and line 82.

Point 5: It is interesting to consider echolocation itself the phenotype - wouldn't it be better the suite of characters contributing to echolocation? The cochlea does make sense here as one representative

Response 5: Thanks for your suggestion. In the revised manuscript, we indicated that resting frequencies (RFs) of horseshoe bats are widely used to characterize echolocation vocalizations in previous studies (Sun et al., 2013, Luo et al., 2019), so we measured RFs in this study. We have added it in lines 115-118.

References:

Luo, B.; Leiser-Miller, L.; Santana, S.E.; Zhang, L.; Liu, T.; Xiao, Y.H.; Liu, Y.; Feng, J., Echolocation call divergence in bats: A. comparative analysis. Behav. Ecol. Sociobiol. 2019, 73, 154.

Sun, K.P.; Luo, L.; Kimball, R.T.; Wei, X.W.; Jin, L.R.; Jiang, T.L.; Feng, J., Geographic variation in the acoustic traits of greater horseshoe bats: Testing the importance of drift and ecological selection in evolutionary processes, PLoS One 2013, 8 (8), e70368

Point 6: Line 108 - intra again

Response 6: We changed “inter-subspecific” to “intraspecific”.

Point 7: Section 2.i - need to mention ethics on the bat collection and their conservation and protection status

Response 7: Thanks for your suggestion. In Section 2.1, We added the ethics statement as “All sampling procedures were approved by National Animal Research Authority in Northeast Normal University, China (approval number: NENU–20080416).” (lines 129-131).

Point 8: Line 162 - a very high level of precision - would suggest to round to one decimal point or to justify this precision - I can see that the differences are not always great but they still do not overlap even at one decimal

Response 8: Based on your suggestion, we rounded the percentage to one decimal point in the revised manuscript.

Point 9: Figure 1 - does the overlap have a meaning? Is it showing it at a scale?

Response 9: The overlap means the number of intersecting DEGs with expression variation forced by natural selection for different organs. To clearly clarify the meaning of this figure, we revised the legend as “Venn diagram showing the numbers of intersecting differentially expressed genes (DEGs) under natural selection among three organs in (A) the inter-specific and (B) inter-subspecific comparison. The labels ‘cochlea’, ‘brain’, and ‘liver’ indicate the cochlea, brain, and liver samples in each comparison, respectively. The size of diagram is proportional to quantity. Colors have no meaning.” (lines 207-211).

Point 10: I am really impressed with the discussion. It is well ordered, puts the work well in the context of the literature, and I find the discussion on the costly immune response of different types of calling in different sexes fascinating and would apply this to my own work. A huge array of references are used putting this work in a broad context.

Response 10: Thank you again for your affirmation and suggestion for our work.
